# Performance of family health teams for tackling chronic diseases in a state of the Amazon

**Kelly Cristina Gomes Alves**[1☯¤a]*, **Rafael Alves Guimarães**[2‡], **Marta Rovery de Souza**[2‡], **Otaliba Libânio de Morais Neto**[2☯¤b]

1 Department of Medicine, Federal University of Tocantins, Palmas, Tocantins, Brazil, 2 Institute of Tropical Pathology and Public Health, Federal University of Goiás, Goiânia, Goiás, Brazil

☯ These authors contributed equally to this work.
¤a Current address: Department of Medicine, Federal University of Tocantins, Palmas, Tocantins, Brazil
¤b Current address: Institute of Tropical Pathology and Public Health, Federal University of Goiás, Goiânia, Goiás, Brazil
‡ These authors also contributed equally to this work
* kellygomesalves@gmail.com

**Data Availability Statement:** The datasets analysed during the current study are available in the Departament of Primary Care (Departamento de Atenção Básica - DAB) repository, https://aps.saude.gov.br/ape/pmaq. To select data from family

## Abstract

The most common cause of death worldwide is noncommunicable diseases. A cross-sectional study was conducted to evaluate the adequacy of the work process among family health teams and compare differences in regional adequacy in the state of Tocantins, in the Amazonian Region, Brazil. Categorical principal components analysis was used, and scores of each principal component extracted in the analysis were compared among health regions in Tocantins. A *post hoc* analysis was performed to compare the heath region pairs. The adequacy of family health teams' work process was evaluated with respect to the Strategic Action Plan to Tackle NCDs. The results showed that the family health teams performed actions according to the Strategic Action Plan to Tackle NCDs. However, overall, the adequacy percentages of these actions according to the axes of the Plan are very uneven in Tocantins, with large variations among health regions. The family health teams in the Bico do Papagaio (Region 1), Médio Norte Araguaia (Region 2), Cantão (Region 4) and Capim Dourado (Region 5) regions have adequacy percentages ≥ 50% with the Strategic Action Plan to Tackle NCDs, whereas all other regions have percentages <50%. Health teams perform surveillance actions, health promotion, and comprehensive care for NCDs in accordance with the guidelines of the Strategic Action Plan to Tackle NCDs. The challenge of NCDs in primary care requires a care model that is tailored to users' needs and has the power to reduce premature mortality and its determinants.

## Introduction

The most common cause of death worldwide is noncommunicable diseases (NCDs), which accounted for 71.0% of deaths in 2016. The main groups of NCDs are cardiovascular diseases, cancer, chronic respiratory diseases and diabetes. These four groups share common determinants, including the individual's socioeconomic conditions and modifiable behavioral factors

health strategy teams: 1. Select Ciclo 2 [Portuguese]; 2. Select Microdados da avaliação Externa [Portuguese]; 3. Then, choose Módulo II Equipe [Portuguese]; 4. Finally, choose Equipe Tocantins [Portuguese] to download the dataset.

**Funding:** The authors received no specific funding for this work.

**Competing interests:** The authors have declared that no competing interests exist.

such as smoking, alcohol abuse, unhealthy diet, physical inactivity and excess weight and obesity [1]. High-income countries have high rates of morbidity and mortality. In low- and middle-income countries, the situation is even more alarming: approximately 86.0% of premature deaths and higher disease burdens and economic losses are due to NCDs [2,3]. In Brazil, almost half of all Brazilians surveyed reported having at least one chronic disease in 2013 [4], and NCDs accounted for 75.8% of all deaths in 2015 [5].

The World Health Organization has taken some initiatives to guide health systems in addressing the strong and growing burden of NCDs worldwide. Some of these initiatives include the Global Strategy for the Prevention and Control of NCDs [6], the Framework Convention on Tobacco Control [7], the Action Plan for the Global Strategy for the Prevention and Control of NCDs [3,8], and the Global Strategy to Reduce the Harmful Use of Alcohol [9]. The 2011 United Nations High-Level Meeting alerted the world to the social and economic impacts of the disease burden, particularly in developing countries [10].

In Brazil, since the implementation of the Unified Health System (*Sistema Único de Saúde —SUS* in Portuguese) in 1988, several initiatives by the Ministry of Health have been proposed to encourage the organization of the health system and the promotion of health for the population [11]. These initiatives have been implemented especially through the improvement of primary health care, a key strategy for addressing NCDs [12,13]. Following the principles and guidelines of the Unified Health System, the Family Health Program was created in 1994 with the aim of redesigning the health care model to promote the quality of care delivered by primary health care. In 2003, the program was officially called the Family Health Strategy [14,15]. The Family Health Strategy consists of the activities of family health teams in primary health units (PHUs–*Unidade Básica de Saúde—UBS* in Portuguese) in clinics for primary care. Family health teams are responsible for offering primary health interventions to the target population, known as the assigned population, comprising 2,000 to 3,500 people who live near the primary health unit.

The family health team comprises at least a doctor, nurse, assistant, or nursing technician and community health agents. Other professionals may be part of the team, such as a dentist [11]. The first point of contact between health services users with the local health system is provided by family health teams. Teams work with coordinated care, integration of care with diagnostics, specialist support, and referral to hospital care according to the needs of health service users. The care is provided in primary health units (PHUs), in patients' homes, and in the community [15]. The team's work process aims to increase treatment and improve the population's health indicators, providing an important cost-effectiveness ratio [11]. Impacts of the Family Health Strategy were observed in the significant reduction in the number of admissions for NCDs [16].

More recent initiatives for strengthening primary care in Brazil include the reformulation in 2011 of the National Primary Care Policy (*Política Nacional de Atenção Básica* in Portuguese), the purpose of which is to promote the expansion of primary care coverage through the Family Health Strategy. This policy also aims to improve NCDs care by providing guidelines for clinical responses, risk stratification and monitoring of chronic conditions. In addition, the policy aims to expand the matrix support from specialists who comprise the Family Health Support Center (*Núcleo Ampliado de Apoio à Saúde da Família—NASF* in Portuguese) to support the decision-making of family health teams [11].

In 2011, the Ministry of Health also launched the Strategic Action Plan to Tackle Noncommunicable Diseases in Brazil 2011–2022. This plan proposed three major areas of intervention: surveillance, information, evaluation and monitoring; health promotion; and comprehensive care. The plan's main goal is a 2.0% annual reduction in premature mortality due to NCDs by 2022 [17]. It has quickly produced results in all three of its intervention areas, meeting targets

in the 2011 to 2015 period, including a reduction in smoking and the promotion of healthy eating and physical activity [18].

Within this framework of expansion and restructuring of primary care in Brazil, the Primary Care Access and Quality Improvement Program (*Programa de Melhoria do Acesso e da Qualidade da Atenção Básica—PMAQ-AB in* Portuguese) was created to improve the quality of the care model and access to primary care. The program is supported by a system that evaluates and monitors the structure of PHUs and the work process of family health teams that are part of the Family Health Strategy [19].

In an attempt to reduce health inequities and increase access, rapid expansion of the Family Health Strategy has taken place in underdeveloped regions of Brazil [20] such as the North and Northeast; however, inequalities persist in these areas [21]. The state of Tocantins recorded the highest proportion of coverage of its population by the Family Health Strategy in the country [21]. Tocantins is located in the North Region, within Amazonia, in an area in which problems in both the structure of the PHUs [22] and in the work process related to the primary care for users with NCDs have been reported [23].

The expansion of coverage and the quality of primary care are especially important for increasing the ability to tackle NCDs [13] in light of the Chronic Care Model (CCM) [24]. Studies that evaluated the CCM [25] have shown improved primary care organization [26], with positive changes in regard to the redesign of health systems, multidisciplinary team performance, professional development strategies for diabetes management and user self-management support. In addition, primary health care has been improved due to the use of clinical protocols and risk stratification, registration of health records in information systems, control of diseases such as diabetes [12] and congestive heart failure [27] and reduced spending on health services [28].

The aim of this study was to evaluate the adequacy of the work process among family health teams according to the National Action Plan to Tackle NCDs and the National Primary Care Policy recommendations and compare regional adequacy differences in the state of Tocantins, Amazonian Region, Brazil. Despite regional differences in the adequacy of the work process, family health teams have advanced in fulfilling the health promotion axes of the plan.

## Materials and methods

### Study design, data source and study population

A cross-sectional study was conducted to evaluate the adequacy and compare regional adequacy differences of actions taken by the Tocantins' family health teams within the intervention areas included in the Strategic Action Plan to Tackle NCDs in Brazil from 2011 to 2022 according to the National Primary Care Policy guidelines.

The state of Tocantins, which includes 139 municipalities, is located in the North Region and is part of Amazonia Region. In 2016, its estimated population was 1,532.902, and it had an annual gross domestic product per capita of 4423,94 dollars. Its main exports are soybeans, meat and meat products. Tocantins has extensive rural areas [29] and has the second largest Indigenous population in the country. In the first two cycles of the PMAQ-AB, it was one of the states with the greatest adherence to the program; 455 family health teams were operating in 135 municipalities in Tocantins in July 2014, covering 91.8% of the population [30]. Of these, 385 (84.6%) health teams in 117 municipalities joined the PMAQ-AB in that year; 361 (79.3%) of these teams were interviewed and are the object of analysis of this study.

In 2012, the state of Tocantins was divided into eight health regions (Fig 1) according to the following inclusion criteria: at least 80.0% coverage by the Family Health Strategy, the presence of an active health surveillance team, and locations for urgent and emergency care, specialized

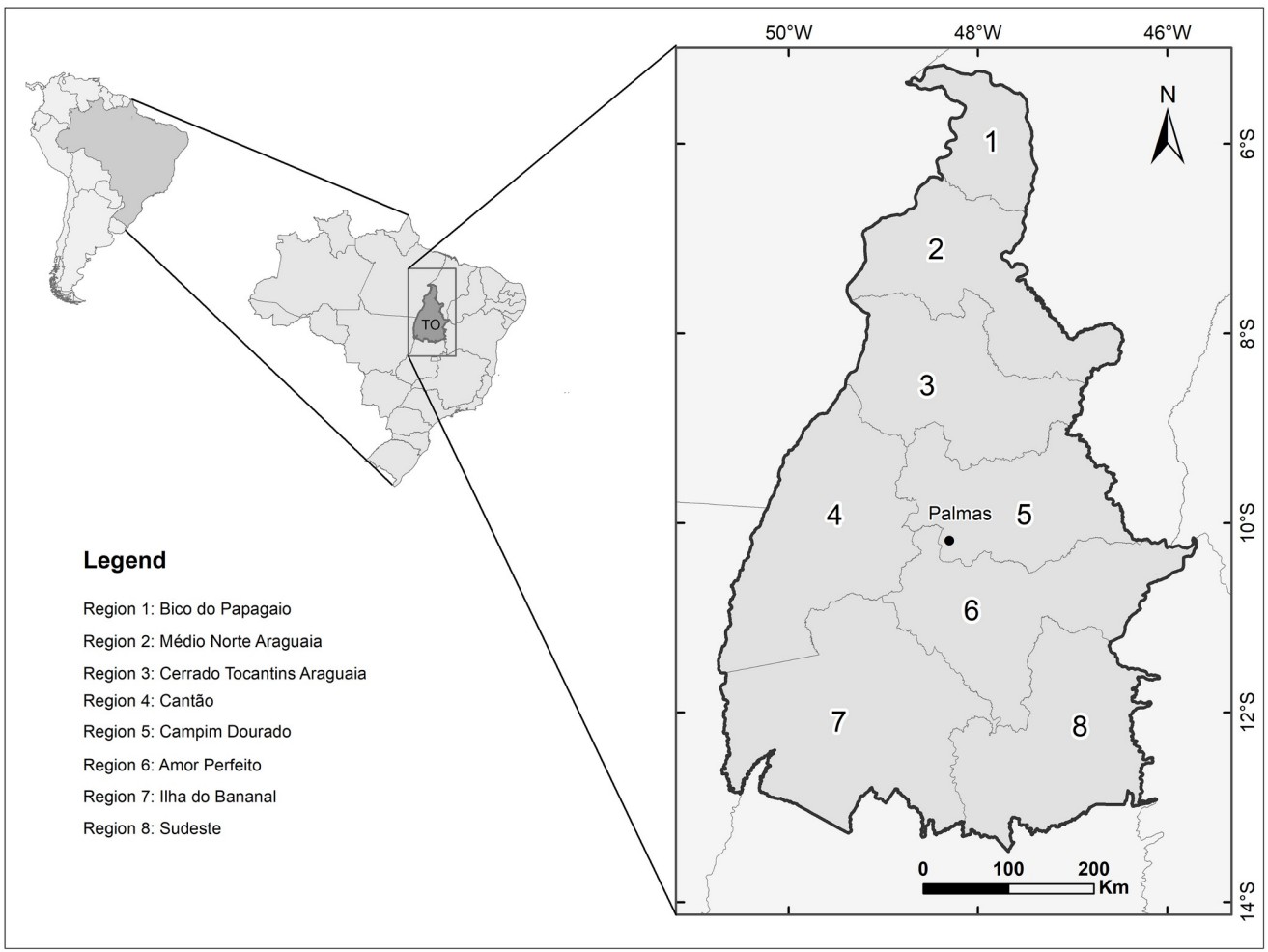

**Fig 1. Health regions, Tocantins, Brazil.**

care, hospital and nonhospital care. Health regions meet one of the principles that guide the organization of the Unified Health System (*Sistema Único de Saúde—SUS* in Portuguese) in Brazil, called the regionalization of health, defined by the Federal Constitution of 1988 and Law 8080/1990 that regulates the Brazilian health system [31]. Currently, the eight health regions of Tocantins are (i) Bico do Papagaio (Region 1), which includes 24 municipalities and has a population of 194,297 inhabitants, corresponding to 13.7% of the population; (ii) Médio Norte Araguaia (Region 2), with 17 municipalities and 269,814 inhabitants (19.0% of the population); (iii) Cerrado Tocantins Araguaia (Region 3), with 23 municipalities and 148,923 inhabitants (10.5% of the population); (iv) Cantão (Region 4), with 15 municipalities and 117,443 inhabitants (8.3% of the population); (v) Capim Dourado (Region 5), with 14 municipalities and 315,621 inhabitants (22.3% of the population); (vi) Amor Perfeito (Region 6), with 13 municipalities and 104,660 inhabitants (7.4% of the population); (vii) Ilha do Bananal (Region 7), with 18 municipalities and 173,586 inhabitants (12.2% of the population); and (viii) Sudeste (Region 8), with 15 municipalities and 93,350 inhabitants (6.6% of the population).

The study's data source was the PMAQ-AB database. This database is currently the largest national database within the Family Health Strategy. It is also an important source of information for evaluating the adequacy of the work process of family health teams with the Strategic Action Plan to Tackle NCDs. The PMAQ-AB has three phases: an adherence and contracting phase (phase 1), a certification phase (phase 2) and a renew contracting phase (phase 3).

Phase 1 consists of the adherence of family health teams to the program. At this stage, municipal management and the family health teams are committed to following the program guidelines and to evaluating and monitoring health indicators. Following this commitment, they sign the agreement with the Ministry of Health.

Phase 2 includes the external evaluation stage in which Family Health Strategy data collection tools, known as modules, are used. The implementation of external evaluation across the country occurred in 2012 (cycle 1), 2014 (cycle 2) and 2017 (cycle 3). Module I of the external evaluation investigated the structure of the PHUs, module II included interviews with the health teams in regard to their work process, and module III included interviews with users to determine their satisfaction with the services received. New modules were added in each PMAQ-AB cycle.

Phase 3 constitutes the agreement of new standards and quality indicators to be achieved by the family health teams committed to the program. Renewed contracting aims to promote cyclical and systematic evaluation processes based on the results achieved by the PMAQ-AB participants [32].

At the time of this study, PMAQ-AB data for the Tocantins were selected from module II from cycle II performed in 2014 because these were the most recent data available. The health team interviews covered their work process and included actions related to management, health promotion, prevention and the comprehensive care of service users.

## Logic model, data collection and variable selection

The logic model (Fig 2) for the evaluation of the adequacy of the Tocantins Family Health Strategy actions with the intervention areas of the Strategic Action Plan to Tackle NCDs had as theoretical references the CCM of Wagner et al. [24] and the evaluation-oriented theoretical model of Hartz and Silva [33,34]. The logic model comprised three intervention areas of the Strategic Action Plan to Tackle NCDs: (i) surveillance, information, evaluation and monitoring; (ii) health promotion; and (iii) comprehensive care of NCDs. The variables selected from

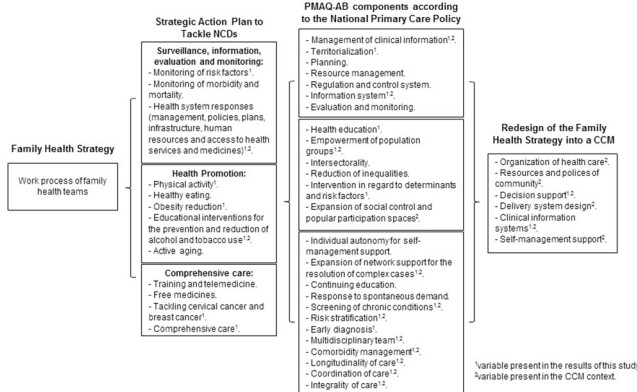

**Fig 2. Logic model evaluating the work process of family health teams according to the PMAQ-AB, 2014.**
Definitions of abbreviations: NCDs = noncommunicable diseases; CCM = chronic care model; PMAQ-AB = Primary Care Access and Quality Improvement Program.

the PMAQ-AB national database concerned the family health teams' work process that corresponded to the three plan areas and were related to the CCM elements (health care organization, service provision system design, decision support, a clinical information system, self-management support and resources and policies of community) [24].

Eighty variables were identified. Those that had no direct relation to the plan's areas and those for which the percentage of missing values was greater than or equal to 10.0% were excluded.

Variables related to the level of training of family health teams were not used because of missing data. However, most teams received training on family medicine for physicians, and family health for nurses, dentists and other professionals of NASF (Family Health Support Center–*Núcleo Ampliado de Apoio à Saúde da Família—NASF* in Portuguese).

This logic model was not subjected to an expert validation process. It was based on models that have been used in published studies [13,17,22–28,33–40], and it was adapted for use with PMAQ-AB data.

## Statistical analysis

First, absolute frequency (n), relative frequency (%) and 95% confidence intervals (95% CIs) were used to describe all the study variables. Second, categorical principal components analysis (CATPCA) was used to identify the variables or group of variables with the greatest explanatory power for the variance between work teams [41,42]. The variables used in this study were categorical; therefore, the use of CATPCA allowed conversion of the variables into quantitative data without loss of the maximum possible variance in the transformation, thus creating a two-dimensional scale [43,44]. This technique is similar to nonlinear principal components analysis (NLPCA) and, together with the use of optimal scaling [43], allowed us to reduce the number of variables and to identify the variables with explanatory and discriminatory power among the family health teams studied [41,42,45].

The first stage in the binary variable factor analysis included the construction of a tetrachoric correlation matrix of all the study variables that could be used to investigate the bivariate relationship between each pair of dichotomous variables [46,47]. This technique replaces the Pearson correlation that is used for quantitative or ordinal variables [45–50]. Variables with correlation coefficients of less than 0.3 ($r_t < 0.3$) with any other matrix variable were excluded from the analysis. The modal value of the respective variable was used for multiple imputation of missing values. The number of principal components extracted by the analysis was determined based on the criteria of an eigenvalue greater than 1 [41,42,51] and Cronbach's alpha $\geq 0.6$ [36]. Each component was constituted using variables with factor loadings $\geq 0.4$ [41,42].

After the identification of the variables that best explain the observed variability, the family health teams' work process and actions were evaluated with respect to their adequacy with the Strategic Action Plan to Tackle NCDs. As used in this study, adequacy is an indicator of delivery of care, from the concept of adequacy inference proposed by Habicht et al. [38]. Component scores are reported as the mean, standard deviation, 95% CI of mean, median, P25, P75 and variance. To compare the scores of the components extracted in the CATPCA between Tocantins health regions, the Kruskal-Wallis test for independent samples was used, and the Dunn *post hoc* test was used in cases of statistical significance. The frequencies and 95% CIs were determined for each variable extracted by CATPCA by health region. Regional differences in adequacy were analyzed using the Pearson chi-square test if all the expected frequencies were greater than five or using Fisher's exact test when at least one expected frequency was less than five [52,53]. When there was a significant (p < 0.05) or marginal (p-value = 0.05 to

0.08) difference, a *post hoc* analysis was performed to compare the pairs. In all analyses, p values < 0.05 were considered statistically significant. CATPCA was performed using the Statistical Package for Social Sciences, version 24.0, and the Kruskal-Wallis, Dunn test, Pearson chi-square test and Fisher's test followed by pairwise comparison were performed using R.

## Results

S1 Table lists and describes the 80 PMAQ-AB variables used in this study to evaluate the family health teams' work process according to the Strategic Action Plan to Tackle NCDs and the National Primary Care Policy recommendations in Tocantins state. These variables are part of the set of health actions that family health teams are expected to perform.

All 80 variables included in the CATPCA had tetrachoric correlation coefficients > 0.3 with at least one selected variable. Analysis of the eigenvalue criterion (eigenvalue > 1) and Cronbach' alpha ≥ 0.6 yielded six principal components (PCs). S2 Table shows the results of the CATPCA. Variables with factor loadings of ≥ 0.4 remained in each PC. Table 1 shows the explanatory variables and their respective factor loadings for each PC defined by the CATPCA method.

The CATPCA organized a set of 80 variables into a scale of 38 two-dimensional variables that can be used in the minimum evaluation of the adequacy of family health teams' work process indicators with the Strategic Action Plan to Tackle NCDs. These 38 variables explain 30.93% of the model variance and have a Cronbach alpha of 0.972. Table 2 indicates the eigenvalue, variance (%), Cronbach's alpha and median scores of principal components.

Table 3 summarizes the scores of each PC retained in the CATPCA for the state of Tocantins and for each health region.

Figs 3 and 4 show *post hoc* analysis of differences in PC scores between health regions.

For PC1 (Management and comprehensive care of users with NCDs), the highest median score was observed in Region 5, which had higher scores than Regions 2, 3, 6, 7 and 8. Region 4 had higher scores than Regions 3 and 8. The lowest score was in Region 2, which had statistically lower scores than Regions 1, 4 and 7. (Fig 3A). For PC2 (Offers care to specific groups and active search), the highest median score was found in Region 2, which had a statistically higher score than Regions 1, 3, 4 and 6. Region 7 had higher scores than Regions 1, 3, and 6. Region 8 had higher scores than Regions 3 and 6. Region 5 had higher scores than Regions 1, 3, and 6. The lowest score was in Regions 3 and 6. (Fig 3B). For PC3 (Diagnostic support and follow-up), Region 1 had a significantly higher score than Regions 3, 5, 7, and 8. Region 7 had a lower score than Regions 2, 3, and 6 (Fig 3C).

For PC4 (Matrix support and electronic health records), the highest score was observed in Region 4. Region 5 had the lowest score, which was significantly lower those in Regions 1, 2, 3, 4, 7 and 8. Region 6 had lower scores than Regions 1, 2 and 4. Finally, Region 4 had a higher score than Region 3 (Fig 4A). No statistically significant difference was found between health regions for PC5 (Health promotion—practice of physical activity) (Fig 4B). For PC6 (Care for users with diabetes), the lowest score was found in Region 1. The score for Region 2 was significantly higher than those for Regions 1, 6 and 8. The score for Region 5 was significantly higher than those for Regions 8 and 1 (Fig 4C).

Table 4 shows the adequacy percentage based on principal components scores for each health region of Tocantins state.

In the analysis by state, the adequacy percentages varied between 48.2% and 54.8%. In the analysis by health regions, Regions 1, 2, 4 and 5 presented at least three PCs with adequacy percentages above 50.0%. The actions most performed by family health teams were those

**Table 1. Factor loading of work process of family health teams (n = 361) captured by the CATPCA from PMAQ-AB.**

| Principal components and variables | Factor loading |
|---|---|
| **PC1 – Management and comprehensive care of users with NCDs** | |
| Offers services to group of users of alcohol and other drugs | 0.421 |
| Offers services to group of users with obesity | 0.523 |
| Offers services to group of users with COPD | 0.466 |
| Has record of users with COPD | 0.449 |
| Has record of women eligible for mammogram | 0.414 |
| Has record of users with obesity | 0.539 |
| Offers consultations for users with obesity | 0.530 |
| Offers consultations for users with COPD | 0.490 |
| Uses protocols for cervical cancer risk stratification | 0.656 |
| Conducts active search for cases of delayed cervical cancer screening | 0.413 |
| Uses protocols for breast cancer risk stratification | 0.666 |
| Uses protocols for hypertension risk stratification | 0.658 |
| Uses protocols for diabetes risk stratification | 0.657 |
| Uses protocols for COPD risk stratification | 0.595 |
| Performs active search for cases of cervical cancer | 0.415 |
| Performs active search for cases of breast cancer | 0.486 |
| Performs active search for cases of hypertension | 0.437 |
| Performs active search for cases of diabetes | 0.448 |
| Performs active search for cases of alcohol and drug use | 0.487 |
| Asks all users about tobacco use | 0.400 |
| **PC2 – Offers care to specific groups and active search** | |
| Offers services to women's groups | -0.575 |
| Offers services to group of elderly users | -0.516 |
| Offers services to group of users with hypertension | -0.659 |
| Offers services to group of users with diabetes | -0.657 |
| Performs active search for cases of delayed cervical cancer screening | 0.420 |
| Performs active search for cases of cervical cancer | 0.465 |
| Performs active search for cases of hypertension | 0.505 |
| Performs active search for cases of diabetes | 0.523 |
| **PC3 – Diagnostic support and follow-up** | |
| Requests creatinine test performed by the service network | 0.520 |
| Requests lipid profile test performed by the service network | 0.445 |
| Requests electrocardiogram performed by the service network | 0.489 |
| Requests glycosylated hemoglobin test performed by the service network | 0.511 |
| Requests mammogram performed by the service network | 0.545 |
| Requests fasting glucose test performed by the service network | 0.464 |
| **PC4 – Matrix support and electronic health records** | |
| Receives matrix support from the NASF to care for people with NCDs | 0.507 |
| Stores electronic medical records on computer | -0.430 |
| Engages the NASF to support the monitoring of obese users in PHUs | 0.460 |
| **PC5 – Health promotion – practice of physical activity** | |
| Encourages and develops physical activity | 0.401 |
| Performs activities in schools to promote physical activity | 0.400 |
| **PC6 – Care for users with diabetes** | |
| Offers consultations for users with diabetes | 0.418 |

(*Continued*)

**Table 1.** (Continued)

| Principal components and variables | Factor loading |
|---|---|
| Works at PHUs that collect blood test | 0.648 |
| Works at PHUs that collect urine test | 0.633 |

Definitions of abbreviations: PC = principal component; COPD = chronic obstructive pulmonary disease; NASF = Family Health Support Center; NCDs = noncommunicable diseases; PHUs = primary health units.

belonging to PC 3 (Diagnostic support and follow-up), PC4 (Matrix support and electronic health records) and PC5 (Health promotion—practice of physical activity).

Table 5 shows the comparative analysis of the family health teams' work process in the Tocantins health regions by PC retained in the CATPCA. PC2 (Offers care to specific groups and active search), PC3 (Diagnostic support and follow-up) and PC5 (Health promotion—practice of physical activity) were practiced by more than 70% of family health teams that performed the recommended actions. Lower percentages (up to 30%) were observed in PC1 (Management and comprehensive care of people with NCDs) and PC4 (Matrix support and electronic health records).

The analysis of the pairwise comparison of the differences between health regions in Tocantins (S3 Table) was able to identify and discriminate between health regions in which the difference was statistically significant. This analysis included 33 of the 38 explanatory variables extracted by the CATPCA because these statistically significant variables remained in the model. The results of the pairwise comparison analysis revealed that the work process of the family health teams in the Bico do Papagaio (Region 1), Cantão (Region 4) and Capim Dourado (Region 5) regions showed higher percentages of recommended actions performed within each PC than those family health teams in the Médio Norte Araguaia (Region 2) and Sudeste (Region 8) regions, which had the lowest PC percentages.

## Discussion

This methodological evaluation of the family health teams performance for tackling NCDs is innovative in the Family Health Strategy in Brazil. Six PCs associated with the adequacy of the work process of family health teams in the Tocantins health regions were identified, and the findings among these regions were compared. It was also possible to identify and evaluate the primary care actions focused on NCDs care that best explained the variability among teams in the regions.

The results showed that the family health teams performed actions according to the Strategic Action Plan to Tackle NCDs. However, overall, the adequacy percentages of these actions according to the axes of the plan are very uneven in Tocantins, with considerable variations in PC adequacy among health regions. Minor variations were observed in the statewide analysis.

**Table 2. Eigenvalue, variance (%), Cronbach's alpha and median scores of principal components retained by CATPCA.**

| Testes | PC1 | PC2 | PC3 | PC4 | PC5 | PC6 |
|---|---|---|---|---|---|---|
| Eigenvalue | 8.731 | 3.859 | 3.697 | 3.041 | 2.751 | 2.675 |
| Variance (%) | 10.91 | 4.82 | 4.62 | 3.80 | 3.44 | 3.34 |
| Cronbach's alpha | 0.897 | 0.750 | 0.739 | 0.680 | 0.644 | 0.634 |
| Scores median | -0.0167 | -0.0690 | 0.0841 | -0.0155 | 0.0516 | 0.0071 |

Definition of abbreviation: PC = principal component.

**Table 3. Percentile of principal component scores in the analysis for each health region and for the state.**

| Principal components | Total | Health regions | | | | | | | |
|---|---|---|---|---|---|---|---|---|---|
| **PC 1** | | 1 | 2 | 3 | 4 | 5 | 6 | 7 | 8 |
| P25 | -0.7160 | -0.3980 | -12.889 | -12.191 | -0.1489 | 0.2761 | -0.7173 | -0.4919 | -11.985 |
| Median | -0.0167 | 0.2308 | -0.8442 | -0.6766 | 0.3432 | 0.8409 | -0.2128 | -0.1058 | -0.2899 |
| P75 | 0.7588 | 0.7952 | -0.2170 | -0.0183 | 0.9145 | 14.989 | 0.4759 | 0.8440 | 0.2466 |
| **PC 2** | | | | | | | | | |
| P25 | -0.6314 | -0.8144 | 0.0327 | -11.330 | -0.6545 | -0.1584 | -11.989 | -0.1415 | -0.2244 |
| Median | -0.0690 | -0.3033 | 0.6775 | -0.8477 | -0.1850 | 0.3696 | -0.8310 | 0.1393 | -0.0552 |
| P75 | 0.4433 | 0.0341 | 17.982 | -0.2356 | 0.3268 | 0.7720 | -0.1071 | 0.3905 | 0.6320 |
| **PC 3** | | | | | | | | | |
| P25 | -0.3226 | 0.0065 | -0.3025 | -0.5432 | -0.1172 | -0.3118 | -0.1792 | -0.9568 | -0.8269 |
| Median | 0.0841 | 0.4140 | 0.2632 | 0.0039 | 0.1997 | -0.0728 | 0.1846 | -0.2424 | -0.1439 |
| P75 | 0.5463 | 0.8155 | 0.5282 | 0.5049 | 0.6869 | 0.4712 | 0.9292 | 0.1302 | 0.1574 |
| **PC 4** | | | | | | | | | |
| P25 | -0.6788 | -0.1772 | -0.3098 | -0.8540 | 0.1264 | -15.640 | -0.9859 | -0.3101 | -0.3807 |
| Median | -0.0155 | 0.3818 | 0.1327 | -0.1817 | 0.4204 | -0.9715 | -0.5811 | 0.1966 | 0.1779 |
| P75 | 0.6584 | 10.068 | 0.7503 | 0.3430 | 10.852 | -0.4001 | 0.1438 | 0.7045 | 0.7996 |
| **PC 5** | | | | | | | | | |
| P25 | -0.6530 | -0.9414 | -0.9802 | -0.3460 | -0.2931 | -0.5437 | -11.966 | -0.6292 | -11.610 |
| Median | 0.0516 | -0.1076 | -0.0243 | 0.1218 | 0.1434 | 0.1620 | -0.4052 | -0.0138 | 0.1302 |
| P75 | 0.6857 | 0.8405 | 0.6108 | 0.9587 | 0.7206 | 0.5010 | 0.1178 | 0.5002 | 0.8963 |
| **PC 6** | | | | | | | | | |
| P25 | -0.6118 | -0.7723 | -0.1016 | -0.6143 | -0.6031 | -0.2346 | -10.837 | -0.8353 | -11.832 |
| Median | 0.0071 | -0.3668 | 0.3463 | -0.1286 | 0.2282 | 0.1724 | -0.3713 | 0.0172 | -0.4365 |
| P75 | 0.6258 | 0.2802 | 0.8473 | 0.6590 | 0.7830 | 0.9688 | 0.1938 | 0.4541 | 0.1065 |

Definitions of abbreviations: PC = principal component; P25 = 25th percentile; P75 = 75th percentile.

Legend of health regions: 1 = Bico de Papagaio; 2 = Médio Norte Araguaia; 3 = Cerrado Tocantins Araguaia: 4 = Cantão: 5 = Capim Dourado; 6 = Amor Perfeito; 7 = Ilha do Bananal; 8 = Sudeste.

The study found that the family health teams in the Bico do Papagaio (Region 1), Médio Norte Araguaia (Region 2), Cantão (Region 4) and Capim Dourado (Region 5) regions have adequacy percentages $\geq$ 50% with the Strategic Action Plan to Tackle NCDs, whereas all other regions have adequacy percentages < 50%. The highest percentages of adequacy were observed in the areas of Diagnostic support and follow-up (PC3), Matrix support and electronic health records (PC4) and Health promotion—practice of physical activity (PC5).

The health teams' actions in the areas of Diagnostic support and follow-up (PC3) and Matrix support and electronic health records (PC4) were consistent with the main National Primary Care Policy guidelines; the assessment included variables relating to requesting tests and matrix support.

The high adequacy percentage in requesting tests contrasts with the low adequacy percentages found for other variables associated with the area of comprehensive care. According to Garnelo [23], a service network's greater ability to offer diagnostic support tests than to follow risk stratification and develop care plan protocols is contradictory, especially when considering that these procedures are inexpensive and feasible within the Family Health Strategy. This finding may indicate a weakness of the health teams in terms of care coordination and longitudinality of care for users with NCDs.

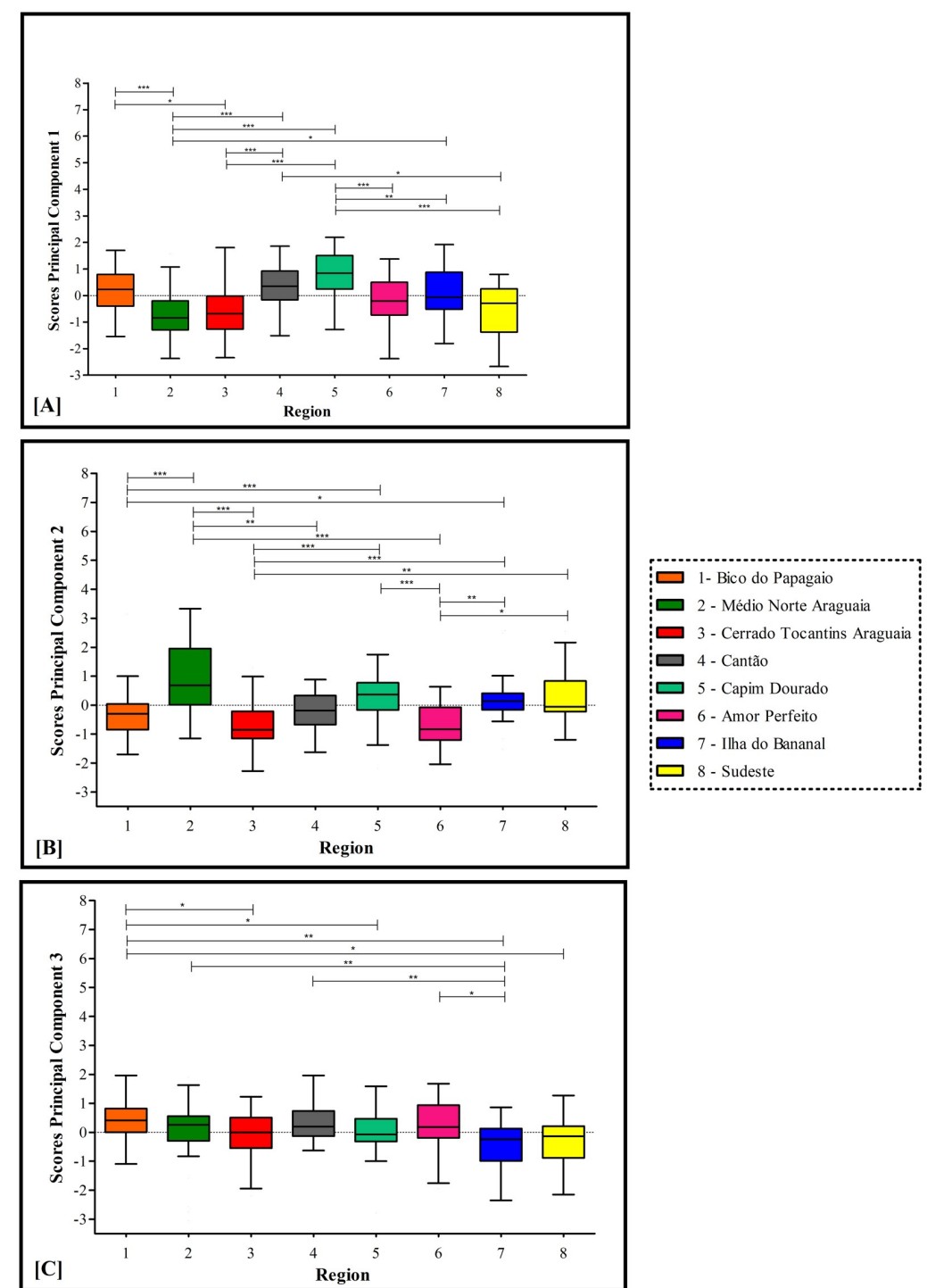

**Fig 3. Principal component scores by health region (PC1 to PC3).** Definition of abbreviation: PC = principal component.

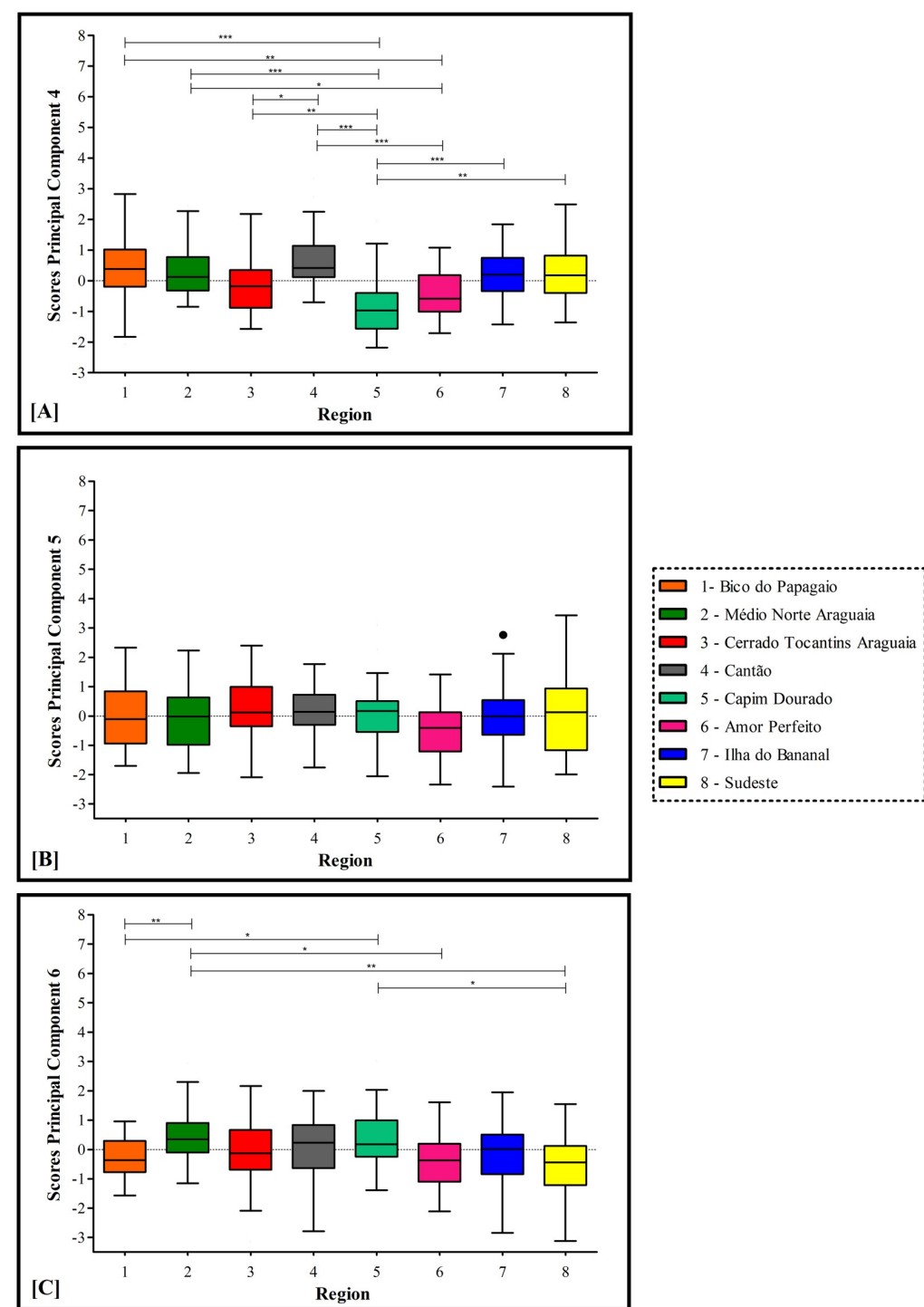

**Fig 4. Principal component scores by health region (PC4 to PC6).** Definition of abbreviation: PC = principal component.

**Table 4. Adequacy percentage based on principal components scores (n = 361).**

| Principal components | n (%) | Health regions | | | | | | | |
|---|---|---|---|---|---|---|---|---|---|
| | | 1 | 2 | 3 | 4 | 5 | 6 | 7 | 8 |
| | | (n = 52) | (n = 53) | (n = 51) | (n = 36) | (n = 70) | (n = 30) | (n = 51) | (n = 18) |
| PC1 | 174(48.2) | 14(26.9) | 40(75.5) | 9(17.6) | 16(44.4) | 47(67.1) | 7(23.3) | 33(64.7) | 8(44.4) |
| PC2 | 180(49.9) | 31(59.6) | 10(18.9) | 12(23.5) | 25(69.4) | 59(84.3) | 13(43.3) | 24(47.1) | 6(33.3) |
| PC3 | 198(54.8) | 39(75.0) | 38(71.7) | 26(51.0) | 23(63.9) | 32(45.7) | 18(60.0) | 16(31.4) | 6(33.3) |
| PC4 | 179(49.6) | 35(67.3) | 30(56.6) | 22(43.1) | 31(86.1) | 8(11.4) | 10(33.3) | 32(62.7) | 11(61.1) |
| PC5 | 191(52.9) | 25(48.1) | 26(49.1) | 32(62.7) | 21(58.3) | 41(58.6) | 10(33.3) | 25(49.0) | 11(61.1) |
| PC6 | 184(51.0) | 19(36.5) | 36(67.9) | 24(47.1) | 22(61.1) | 40(57.1) | 12(40.0) | 26(51.0) | 5(27.8) |

Definitions of abbreviations: PC = principal component; NCDs = noncommunicable diseases; PHUs = primary health units.

Legend of health regions: 1 = Bico de Papagaio; 2 = Médio Norte Araguaia; 3 = Cerrado Tocantins Araguaia: 4 = Cantão: 5 = Capim Dourado; 6 = Amor Perfeito;

7 = Ilha do Bananal; 8 = Sudeste.

The family health teams received matrix support in almost half of family health teams of the state, with large variation in adequacy among health regions. Studies of the NASF conducted in Brazil using PMAQ-AB data indicate that the lowest proportion of matrix support is found in the North Region [54,55] and corroborate the findings for Tocantins [55]. Matrix support was associated with better performance of health teams, work process improvements, comprehensive care, and Family Health Strategy organization [55]. This support promotes the proficiency of family health team members through interaction with NASF professionals in dealing with more complex clinical cases, obtaining second opinions from specialists, diagnosis, and management in such cases [56]. Matrix support may be considered an important marker of adequacy with the Strategic Action Plan to Tackle NCDs and with the National Primary Care Policy guidelines.

Regarding the Health promotion—practice of physical activity (PC5), this study showed higher adequacy percentages than that observed in a study conducted in Brazil in 2009 [57]. Actions that promote physical activity, including offering public spaces and facilities for physical activity, have been effective in terms of interventions focused on individuals, their families and their environment [58–60]. However, advances in intersectoral action for more effective health promotion represent a great challenge for society [59].

In the surveillance area, including health actions for the surveillance of risk factors, the family health teams performed several strategic actions proposed by the Strategic Action Plan to Tackle NCDs; however, lower adequacy percentages were identified for this intervention area. These findings illustrate the need for more investment in actions involving obesity control and alcohol abuse monitoring. Actions aimed at preventing the use of alcohol and other drugs that occurred in less than half of PHUs in Brazil in 2009 [57] continue to represent a challenge in reducing these risk factors.

The variables that were notable in terms of low adequacy in the surveillance area were the recording of user information (diabetes, obesity and COPD) and the use of electronic medical records in all health regions. Lack of recording of user information and low electronic medical record use compromise risk factor surveillance, care coordination, and longitudinality [61]. Moreover, these limitations highlighted structural deficiencies in the PHUs and reflect the lack of IT incorporation. This condition is present not just in Tocantins but across the country [62]. The incorporation of IT in the Family Health Strategy can lead to improvements in primary care organization, evidence-based policymaking [63], performance of health teams and

**Table 5. Comparative analysis of the family health teams' work process (n = 361) by PC retained in the CATPCA for each health region.**

| Principal components and variables | n (%) | Health regions | | | | | | | | p-value |
|---|---|---|---|---|---|---|---|---|---|---|
| | | 1 | 2 | 3 | 4 | 5 | 6 | 7 | 8 | |
| | | (n = 52) | (n = 53) | (n = 51) | (n = 36) | (n = 70) | (n = 30) | (n = 51) | (n = 18) | |
| **PC1** | | | | | | | | | | |
| Offers services to group of users of alcohol and other drugs | 140 (38.8) | 31 (59.6) | 9 (17.0) | 15 (39.4) | 13 (36.1) | 34 (48.6) | 13 (43.3) | 19 (37.3) | 6 (33.3) | 0.001* |
| Offers services to group of users with obesity | 161 (44.6) | 32 (61.5) | 12 (22.6) | 23 (45.1) | 19 (52.8) | 35 (50.0) | 14 (46.7) | 18 (35.3) | 8 (44.4) | 0.006* |
| Offers services to group of users with COPD | 133 (36.8) | 23 (44.2) | 8 (15.1) | 17 (33.3) | 15 (41.7) | 32 (45.7) | 12 (40.0) | 21 (41.2) | 5 (27.8) | 0.024* |
| Has record of users with COPD | 61 (16.9) | 2 (3.8) | 6 (11.3) | 7 (13.7) | 8 (22.2) | 22 (31.4) | 4 (13.3) | 12 (23.5) | - | < 0.001† |
| Has record of women eligible for mammogram | 111 (30.7) | 7 (13.50) | 8 (15.1) | 11 (21.6) | 12 (33.3) | 34 (48.6) | 8 (26.7) | 25 (29.0) | 6 (33.3) | < 0.001* |
| Has record of users with obesity | 61 (16.9) | 2 (3.8) | 5 (9.4) | 5 (9.8) | 8 (22.2) | 23 (32.9) | 4 (13.3) | 12 (23.5) | 2 (11.1) | 0.001† |
| Offers consultations for users with obesity | 169 (46.8) | 24 (46.2) | 11 (20.8) | 16 (31.4) | 29 (80.6) | 48 (68.6) | 14 (26.7) | 21 (41.2) | 6 (33.3) | < 0.001* |
| Offers consultations for users with COPD | 148 (41.0) | 23 (44.2) | 7 (23.2) | 15 (29.4) | 25 (69.4) | 38 (54.3) | 13 (43.4) | 22 (43.1) | 5 (27.8) | < 0.001* |
| Uses protocols for cervical cancer risk stratification | 200 (55.4) | 36 (69.2) | 15 (28.3) | 15 (29.4) | 24 (66.7) | 61 (87.1) | 14 (46.7) | 27 (54.9) | 7 (38.9) | < 0.001* |
| Conducts active search for cases of delayed cervical cancer screening | 228 (63.2) | 27 (51.9) | 36 (67.9) | 20 (39.2) | 24 (66.7) | 47 (67.1) | 14 (46.7) | 47 (92.2) | 13 (72.2) | < 0.001* |
| Uses protocols for breast cancer risk stratification | 161 (44.6) | 28 (53.8) | 10 (18.9) | 12 (23.5) | 24 (66.7) | 50 (71.4) | 13 (43.3) | 19 (37.3) | 5 (27.8) | < 0.001* |
| Uses protocols for hypertension risk stratification | 184 (51.0) | 36 (69.2) | 15 (28.3) | 13 (25.5) | 18 (50.0) | 57 (81.4) | 14 (46.7) | 25 (49.0) | 6 (33.3) | < 0.001* |
| Uses protocols for diabetes risk stratification | 184 (51.0) | 33 (63.5) | 15 (28.3) | 13 (25.5) | 19 (52.8) | 58 (82.9) | 15 (50.0) | 25 (49.0) | 6 (33.3) | < 0.001* |
| Uses protocols for COPD risk stratification | 101 (28.0) | 18 (34.6) | 5 (9.4) | 6 (11.8) | 11 (30.6) | 40 (57.1) | 7 (23.3) | 13 (25.5) | 1 (5.6) | < 0.001* |
| Performs active search for cases of cervical cancer | 243 (67.3) | 29 (55.8) | 41 (77.4) | 22 (43.1) | 25 (69.4) | 54 (77.1) | 12 (40.0) | 48 (94.1) | 12 (66.7) | < 0.001* |
| Performs active search for cases of breast cancer | 181 (50.1) | 24 (46.2) | 25 (47.2) | 15 (29.4) | 22 (61.1) | 42 (60.0) | 11 (36.7) | 32 (62.7) | 10 (55.6) | 0.007* |
| Performs active search for cases of hypertension | 265 (73.4) | 40 (76.9) | 44 (83.0) | 21 (41.2) | 27 (75.0) | 55 (78.6) | 17 (56.7) | 48 (94.1) | 13 (72.2) | < 0.001† |
| Performs active search for cases of diabetes | 259 (71.7) | 38 (73.1) | 44 (83.0) | 21 (41.2) | 25 (69.4) | 55 (78.6) | 15 (50.0) | 48 (94.1) | 13 (72.2) | < 0.001* |
| Performs active search for cases of alcohol and drug use | 85 (23.5) | 13 (25.0) | 9 (17.0) | 5 (9.8) | 12 (33.3) | 22 (31.4) | 5 (16.7) | 16 (31.4) | 3 (16.7) | 0.042† |
| Asks all users about tobacco use | 150 (41.6) | 25 (48.1) | 19 (35.8) | 14 (27.5) | 18 (50.0) | 46 (65.7) | 11 (36.7) | 12 (23.5) | 5 (27.8) | < 0.001* |
| **PC2** | | | | | | | | | | |
| Offers services to women's groups | 416 (87.5) | 49 (94.2) | 30 (56.6) | 49 (96.1) | 35 (97.2) | 64 (91.4) | 30 (100.0) | 47 (92.2) | 12 (66.7) | < 0.001† |
| Offers services to group of elderly users | 326 (90.3) | 50 (96.2) | 36 (67.9) | 48 (94.1) | 35 (97.2) | 65 (92.9) | 30 (100.0) | 48 (94.1) | 14 (77.8) | < 0.001† |
| Offers services to group of users with hypertension | 335 (92.8) | 51 (98.1) | 37 (69.8) | 49 (96.1) | 36 (100.0) | 69 (98.6) | 30 (100.0) | 49 (96.1) | 14 (77.8) | < 0.001† |
| Offers services to group of users with diabetes | 333 (92.2) | 50 (96.2) | 36 (67.9) | 49 (96.1) | 36 (100.0) | 69 (98.6) | 30 (100.0) | 49 (96.1) | 14 (77.8) | < 0.001† |
| Performs active search for cases of delayed cervical cancer screening | 228 (63.2) | 27 (51.9) | 36 (67.9) | 20 (39.2) | 24 (66.7) | 47 (67.1) | 14 (46.7) | 47 (92.2) | 13 (72.2) | < 0.001* |
| Performs active search for cases of cervical cancer | 243 (67.3) | 29 (55.8) | 41 (77.4) | 22 (43.1) | 25 (69.4) | 54 (77.1) | 12 (40.0) | 48 (94.1) | 12 (66.7) | < 0.001* |
| Performs active search for cases of hypertension | 265 (73.4) | 40 (76.9) | 44 (83.0) | 21 (41.2) | 27 (75.0) | 55 (78.6) | 17 (56.7) | 48 (94.1) | 13 (72.2) | < 0.001† |
| Performs active search for cases of diabetes | 259 (71.7) | 38 (73.1) | 44 (83.0) | 21 (41.2) | 25 (69.4) | 55 (78.6) | 15 (50.0) | 48 (94.1) | 13 (72.2) | < 0.001* |
| **PC3** | | | | | | | | | | |
| Requests creatinine test performed by the service network | 354 (98.1) | 51 (98.1) | 51 (96.2) | 50 (98.0) | 36 (100.0) | 69 (98,6) | 29 (96.7) | 50 (98.0) | 18 (100.0) | 0.962† |
| Requests lipid profile test performed by the service network | 345 (95.6) | 50 (96.2) | 50 (94.3) | 47 (92.2) | 35 (97.2) | 69 (98.6) | 29 (96.7) | 48 (94.1) | 17 (94.4) | 0.750† |
| Requests electrocardiogram performed by the service network | 327 (90.6) | 50 (96.2) | 49 (92.5) | 35 (68.6) | 36 (100.0) | 68 (97.1) | 27 (90.0) | 45 (88.2) | 17 (94.4) | < 0.001† |
| Requests glycosylated hemoglobin test performed by the service network | 330 (91.4) | 49 (94.2) | 47 (88.7) | 47 (92.2) | 35 (97.2) | 68 (97.1) | 27 (90.0) | 43 (84.3) | 14 (77.8) | 0.062† |
| Requests mammogram performed by the service network | 337 (93.4) | 52 (100.0) | 48 (90.6) | 41 (80.4) | 36 (100.0) | 68 (97.1) | 27 (90.0) | 47 (92.2) | 18 (100.0) | < 0.001† |
| Requests fasting glucose test performed by the service network | 359 (99.4) | 52 (100.0) | 52 (98.1) | 51 (100.0) | 36 (100.0) | 70 (100.0) | 29 (96.7) | 51 (100.0) | 18 (100.0) | 0.316† |
| **PC4** | | | | | | | | | | |
| Receives matrix support from the NASF to care for people with NCDs | 147 (40.7) | 38 (73.1) | 20 (37.7) | 16 (31.4) | 28 (77.8) | 3 (4.3) | 15 (50.0) | 21 (41.2) | 6 (33.3) | < 0.001* |
| Stores electronic medical records on computer | 38 (10.5) | 1 (1.9) | 2 (3.8) | 2 (3.9) | 1 (2.8) | 32 (45.7) | - | - | - | <0.001† |
| Engages the NASF to support the monitoring of obese users in PHUs | 158 (43.8) | 35 (67.3) | 27 (50.9) | 18 (35.3) | 27 (75.0) | 16 (22.9) | 15 (50.0) | 16 (31.4) | 4 (22.2) | < 0.001* |
| **PC5** | | | | | | | | | | |
| Encourages and develops physical activity | 247 (70.4) | 43 (82.7) | 36 (69.2) | 39 (83.0) | 31 (86.1) | 51 (72.9) | 8 (28.6) | 30 (58.8) | 9 (60.0) | <0.001† |
| Performs activities in schools to promote physical activity | 208 (63.4) | 34 (68.0) | 32 (64.0) | 29 (65.9) | 27 (77.1) | 33 (57.9) | 12 (42.9) | 34 (69.4) | 7 (46.7) | 0.117† |
| **PC6** | | | | | | | | | | |
| Offers consultations for users with diabetes | 353 (97.8) | 51 (98.1) | 53 (100.0) | 50 (98.0) | 34 (94.4) | 69 (98.6) | 30 (100.0) | 50 (98.0) | 16 (88.9) | 0.157† |
| Works at PHUs that collect blood test | 153 (42.4) | 16 (30.8) | 39 (73.6) | 15 (29.4) | 21 (58.3) | 34 (48.6) | 7 (23.3) | 18 (35.3) | 3 (16.7) | < 0.001* |

(*Continued*)

**Table 5.** (*Continued*)

| Principal components and variables | n (%) | Health regions | | | | | | | | |
|---|---|---|---|---|---|---|---|---|---|---|
| | | 1 | 2 | 3 | 4 | 5 | 6 | 7 | 8 | p-value |
| | | (n = 52) | (n = 53) | (n = 51) | (n = 36) | (n = 70) | (n = 30) | (n = 51) | (n = 18) | |
| Works at PHUs that collect urine test | 149 (41.3) | 16 (30.8) | 39 (73.6) | 13 (25.5) | 21 (58.3) | 36 (51.4) | 7 (23.3) | 14 (27.5) | 3 (16.7) | < 0.001* |

Definitions of abbreviations: PC = principal component; COPD = chronic obstructive pulmonary disease; NASF = Family Health Support Center;

NCDs = noncommunicable diseases; PHUs = primary health units.

Definitions of symbols:

\* = Pearson's chi-square test;

† = Fisher's exact test.

Legend of health regions: 1 = Bico de Papagaio; 2 = Médio Norte Araguaia; 3 = Cerrado Tocantins Araguaia: 4 = Cantão: 5 = Capim Dourado; 6 = Amor Perfeito;

7 = Ilha do Bananal; 8 = Sudeste.

the care for NCDs [64,65], and it may help in decision support and in the implementation of clinical information systems [28]. In Brazil, a recent initiative coordinated by the Ministry of Health involving telehealth and the implementation of e-SUS (electronic health information system of the Unified Health System–*Sistema Único de Saúde—SUS* in Portuguese) may promote patient monitoring and adherence to clinical protocols [66] by the family health teams.

In relation to the primary care model for NCDs, our findings reveal the Family Health Strategy's lack of power in terms of comprehensive care for NCDs in light of the CCM in Tocantins, represented by PC1. The variables related to the use of clinical protocols, user information recording and use of electronic medical records were the closest to the CCM elements, such as decision support and clinical information systems. However, the results of this study revealed low percentages of adequacy for these variables. Problems related to a lack of proficiency in user registration and the use of electronic medical records are a barrier to the implementation of a chronic disease care model [67] along the lines of the CCM.

Although the coverage of the Family Health Strategy has been associated with a decrease in health inequities and admissions due to NCDs [16,21], there are still challenges for tackling NCDs in primary health care, such as fragmented organization care, limited use of protocols, and incipient longitudinal and integral care [22,23,39], which corroborate our findings.

The regional differences observed in this study indicate better adequacy percentages in health regions 1 (Bico do Papagaio), 2 (Médio Norte Araguaia), 4 (Cantão) and 5 (Capim Dourado), suggesting that these regions may be geographically positioned in a way that offers greater and better access to health services. Health teams that are concentrated in geographic areas that have higher numbers of primary care points were previously shown to produce the best work performance in terms of PMAQ-AB certification [68]. Studies using PMAQ-AB data have identified greater adequacy in terms of both the structure and the work process of health teams in municipalities with higher populations and higher human development indices [20,22,39,40].

Another possible explanation for the regional differences observed in this study may lie in the fact that the Bico do Papagaio (Region 1) borders two populous cities, one of which is in Region 2 (Médio Norte Araguaia), that have a number of primary care, specialized and hospital care facilities and serve as health care referral centers for neighboring states. The same can be said of Cantão (Region 4), which includes several municipalities concentrated in areas neighboring Capim Dourado (Region 5), where the capital city of Palmas, which provides the main referral health services for the entire state, is located. It is possible that health regions

more distant from the regions with the highest concentrations of health services and regional referral centers tend to exhibit lower adequacy with the Family Health Strategy due to a lack of structured services.

There are a number of limitations to this study. Of particular concern is that the voluntary adherence to PMAQ-AB, which is associated with financial incentives, may have caused better-performing family health teams to participate in the program. However, this limitation may have been minimized by the high percentage of adherence among the Tocantins health teams. The PMAQ-AB database provides secondary data and is not designed specifically to evaluate actions focused on NCDs. In addition, the external evaluation instrument was not designed to evaluate the major intervention areas of the Strategic Action Plan to Tackle NCDs and the components of the CCM. However, these limitations may have been minimized with the logic model designed for this study, which is capable of selecting those variables that were directly related to primary care for NCDs. Furthermore, the proposed evaluation of the Family Health Strategy from PMAQ-AB data is innovative and has the potential to improve both the program and the evaluation of the Family Health Strategy in Brazil.

It is necessary to improve the current care model by introducing better technology in the PHUs, especially in regard to electronic health records. The use of care protocols focused on the management of NCDs, an interdisciplinary approach, matrix support and the coordination of individual and collective care actions are fundamental to the adequate care model. It is important to extend screening actions beyond cervical and breast cancer to cover other NCDs and to reduce barriers to access to primary care for certain population groups such as men and people in socially vulnerable situations. Health education actions should target interventions that address the social determinants of health that operate beyond the PHUs, in particular, intersectoral actions for health promotion. Such measures will increase the Family Health Strategy's power to provide care to the population and may contribute to the fulfillment of the Strategic Action Plan to Tackle NCDs' goals in the state of Tocantins.

## Conclusion

Despite regional differences in the adequacy of work process and actions, the family health teams in Tocantins are making progress in fulfilling the major intervention areas proposed in the Strategic Action Plan to Tackle NCDs. This progress has occurred especially in the health promotion area by means of actions directed at specific groups, including women and the elderly and patients with hypertension and diabetes, and actions that encourage physical activity. The health teams in Regions 1, 2, 4 and 5 had the highest percentages of adequacy in regard to the plan's health promotion area and the lowest percentages for the surveillance area.

The study identified the lack of an NCDs treatment care model in primary care, as recommended by CCM, as a major weakness of the family health teams. While health teams perform surveillance actions, health promotion and comprehensive care for NCDs according to the guidelines of the Strategic Action Plan to Tackle NCDs, the challenge of NCDs in primary care requires a care model that is tailored to users' needs and that has the power to reduce premature mortality and its determinants.

## Supporting information

**S1 Table. Descriptive analysis of the work process variables of family health teams (n = 361) used in the study.** PMAQ-AB Cycle 2, Tocantins, Northern Brazil. Definitions of abbreviations: CI = confidence interval; PMAQ-AB = primary care access and quality improvement program; NASF = family health support center; NCDs = noncommunicable diseases; SIAB = primary care information system; e-SUS = Unified Health System electronic

information; COPD = chronic obstructive pulmonary disease; PHUs = primary health units.
(DOCX)

**S2 Table. Factor loading of variables of work process of family health teams captured by the CATPCA method.** PMAQ-AB Cycle 2, Tocantins, Northern Brazil. Definitions of abbreviations: CI = confidence interval; PMAQ-AB = primary care access and quality improvement program; NASF = family health support center; NCDs = noncommunicable diseases; SIAB = primary care information system; e-SUS = Unified Health System electronic information; COPD = chronic obstructive pulmonary disease; PHUs = primary health units. Definitions of symbols: † = Factor loading ≥ 0.4.
(DOCX)

**S3 Table. Pairwise comparison analysis of the differences between the health regions, Tocantins, Northern Brazil.** Definitions of abbreviations: COPD = chronic obstructive pulmonary disease; NASF = Family Health Support Center; NCDs = noncommunicable diseases; PHUs = primary health units.
(DOCX)

## Acknowledgments

The authors would like to thank Professor João Ricardo Nickenig Vissoci for reading and commenting on the first version of this manuscript.

## Author Contributions

**Conceptualization:** Kelly Cristina Gomes Alves, Marta Rovery de Souza, Otaliba Libânio de Morais Neto.

**Data curation:** Kelly Cristina Gomes Alves.

**Formal analysis:** Kelly Cristina Gomes Alves, Rafael Alves Guimarães.

**Investigation:** Kelly Cristina Gomes Alves.

**Methodology:** Kelly Cristina Gomes Alves, Rafael Alves Guimarães, Marta Rovery de Souza, Otaliba Libânio de Morais Neto.

**Supervision:** Marta Rovery de Souza, Otaliba Libânio de Morais Neto.

**Visualization:** Kelly Cristina Gomes Alves.

**Writing – original draft:** Kelly Cristina Gomes Alves, Rafael Alves Guimarães, Otaliba Libânio de Morais Neto.

**Writing – review & editing:** Kelly Cristina Gomes Alves, Rafael Alves Guimarães, Marta Rovery de Souza, Otaliba Libânio de Morais Neto.

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
