## [Decision Letter · Decision Letter 0]

15 Jun 2020

PONE-D-19-35881

Performance of the family health teams for tackling chronic diseases in a state of the Amazon.

PLOS ONE

Dear Dr. Gomes Alves,

Thank you for submitting your manuscript to PLOS ONE. After careful consideration, we feel that it has merit but does not fully meet PLOS ONE’s publication criteria as it currently stands. Therefore, we invite you to submit a revised version of the manuscript that addresses the points raised during the review process.

Please address the changes recommended by the reviewers. In addition, please address the following: 

Page 13: Lines 110-111 - Please clarify the following statement: “The state is divided into eight health regions (Fig. 1) that were instituted after meeting the following inclusion criteria…

I would assume that the health regions exist despite the study, so not sure how the inclusion criteria were used to develop the 8 regions?

Page 14: Lines 130-131 - Can you describe these 3 phases?

Page 15: Line 144- logical should be changed to logic I believe.

As noted by both reviewers, I would also strongly recommend that you include more information on the strategy itself as well as a description of the Family Health Teams. 

Furthermore, I would ask that you carefully review the manuscript for minor edits such as spelling, use of words, and other errors. 

We look forward to receiving your revised manuscript.

Kind regards,

Nelly Oelke

Academic Editor

PLOS ONE

Journal Requirements:

Reviewers' comments:

Reviewer's Responses to Questions

**Comments to the Author**

1. Is the manuscript technically sound, and do the data support the conclusions?

Reviewer #1: Yes

Reviewer #2: Yes

2. Has the statistical analysis been performed appropriately and rigorously? 

Reviewer #1: Yes

Reviewer #2: Yes

3. Have the authors made all data underlying the findings in their manuscript fully available?

Reviewer #1: Yes

Reviewer #2: Yes

4. Is the manuscript presented in an intelligible fashion and written in standard English?

Reviewer #1: Yes

Reviewer #2: Yes

5. Review Comments to the Author

Reviewer #1: This is an important evaluation of the performance of the Brazilian primary care in light of the Chronic Care Model and recent national reforms aimed at controlling non-communicable diseases. The study utilized existing data and subjected it to a robust analysis.

A minor limitation in the report is a lack of detail in describing the "Family Health Strategy" and the "Family Health Teams". These are the focus of the research and warrant more detailed description since these are not generic to other health systems. Currently the discussion is of a higher order and does not adequately describe these well enough for readers not familiar with the Brazilian primary care reforms. The principal components which constitute the data gives the reader a sense of what Family Health Teams do. However, other details such as the level of training (cadre) about these health care providers are lacking. It would be worth citing literature about Family Health Teams.

A few typographical errors are present in the manuscript and would need attention through further proofreading.

Reviewer #2: The aim of the study was to evaluate the adequacy of the work process among family health teams and to compare regional adequacy differences in the state of Tocantins, in Amazonian Region, Brazil. A cross-sectional study was conducted to evaluate the adequacy of the work process among family health teams and compare regional adequacy differences in the state of Tocantins, in Amazonian Region, Brazil. It was carried out a cross-sectional using the PMAQ-AB secondary database. The method is described with rigor and details. The results and discussions answered adequately the study’s aim. The methodology evaluation of the family health teams for tackling NCDs is innovative in the Family Health Strategy, in Brazil. It was possible to determine six PC associated with the family health teams' work process adequacy in the Tocantins health regions and to compare the findings among these regions.

My recommendation for the authors is to include in the introduction more information related to the work process and the composition of the family health teams. It is important because the variables selected from the PMAQ-AB national database concerned the family health teams' work process. In addition, it is important to highlight the specifics of work process and family health teams in primary care in Brazil.

6. PLOS authors have the option to publish the peer review history of their article (what does this mean?). If published, this will include your full peer review and any attached files.

Reviewer #1: No

Reviewer #2: Yes: José Luís Guedes dos Santos

---

## [Author Response · Author response to Decision Letter 0]

22 Aug 2020

Page 13: Lines 110-111 - Please clarify the following statement: “The state is divided into eight health regions (Fig. 1) that were instituted after meeting the following inclusion criteria…

I would assume that the health regions exist despite the study, so not sure how the inclusion criteria were used to develop the 8 regions?

Page 14: Lines 130-131 - Can you describe these 3 phases?

Page 15: Line 144- logical should be changed to logic I believe.

As noted by both reviewers, I would also strongly recommend that you include more information on the strategy itself as well as a description of the Family Health Teams. 

Furthermore, I would ask that you carefully review the manuscript for minor edits such as spelling, use of words, and other errors. 

Reviewer #1: This is an important evaluation of the performance of the Brazilian primary care in light of the Chronic Care Model and recent national reforms aimed at controlling non-communicable diseases. The study utilized existing data and subjected it to a robust analysis.

A minor limitation in the report is a lack of detail in describing the "Family Health Strategy" and the "Family Health Teams". These are the focus of the research and warrant more detailed description since these are not generic to other health systems. Currently the discussion is of a higher order and does not adequately describe these well enough for readers not familiar with the Brazilian primary care reforms. The principal components which constitute the data gives the reader a sense of what Family Health Teams do. However, other details such as the level of training (cadre) about these health care providers are lacking. It would be worth citing literature about Family Health Teams.

A few typographical errors are present in the manuscript and would need attention through further proofreading.

Reviewer #2: The aim of the study was to evaluate the adequacy of the work process among family health teams and to compare regional adequacy differences in the state of Tocantins, in Amazonian Region, Brazil. A cross-sectional study was conducted to evaluate the adequacy of the work process among family health teams and compare regional adequacy differences in the state of Tocantins, in Amazonian Region, Brazil. It was carried out a cross-sectional using the PMAQ-AB secondary database. The method is described with rigor and details. The results and discussions answered adequately the study’s aim. The methodology evaluation of the family health teams for tackling NCDs is innovative in the Family Health Strategy, in Brazil. It was possible to determine six PC associated with the family health teams' work process adequacy in the Tocantins health regions and to compare the findings among these regions.

My recommendation for the authors is to include in the introduction more information related to the work process and the composition of the family health teams. It is important because the variables selected from the PMAQ-AB national database concerned the family health teams' work process. In addition, it is important to highlight the specifics of work process and family health teams in primary care in Brazil.

---

## [Editor Report · Decision Letter 1]

2 Sep 2020

PONE-D-19-35881R1

Performance of family health teams for tackling chronic diseases in a state of the Amazon.

PLOS ONE

Dear Dr. Gomez Alves,

Thank you for submitting your manuscript to PLOS ONE. After careful consideration, we feel that it has merit but does not fully meet PLOS ONE’s publication criteria as it currently stands. Therefore, we invite you to submit a revised version of the manuscript that addresses the points raised during the review process.

Please attend to the minor revisions requested. All revisions are required for consideration of this manuscript for publication. 

We look forward to receiving your revised manuscript.

Kind regards,

Nelly Oelke

Academic Editor

PLOS ONE

Additional Editor Comments (if provided):

Minor revisions requested:

• Final sentence in the abstract is not clear. As it currently stands, it does not make sense. Please revise. Would also suggest breaking up into two sentences.

• Line 67, delete “in order.” Not necessary.

• Lines 80-81, “reduce the population's health indicators” is not clear.

• Line 117, remove the first “the.”

• Line 137, suggest capitalizing “Indigenous.”

• Line 169, “adherence to family health,” change the “to” to “of”

• Line 210: “other NASF professionals nonphysicians,” this is not clear. Also need to spell out acronym.

• Review manuscript for consistency in capitalization of “Regions” when referring to a specific numbered region. They should all be capitalized.

---

## [Author Response · Author response to Decision Letter 1]

15 Oct 2020

RESPONSE TO EDITOR:

Comment:

Dear Dr. Gomes Alves,

Thank you for submitting your manuscript to PLOS ONE. After careful consideration, we feel that it has merit but does not fully meet PLOS ONE’s publication criteria as it currently stands. Therefore, we invite you to submit a revised version of the manuscript that addresses the points raised during the review process.

Please attend to the minor revisions requested. All revisions are required for consideration of this manuscript for publication.

Response from the authors: Thank you for your careful review and comments on our manuscript submitted to PlosOne. We are submitting the manuscript which was reviewed according to Editor’ comments and recommendations. In addition, this letter presents the responses of every note made by the Editor. The manuscript was revised with track changes. We certify that our article meets the standards of the journal. In addition, the manuscript was revised in English by the American Journal Experts.

Comment: Final sentence in the abstract is not clear. As it currently stands, it does not make sense. Please revise. Would also suggest breaking up into two sentences.

Response from the authors: Thank you for your careful review. We broke up the final sentence into two sentences to make it more clear.

Comment: Line 67, delete “in order.” Not necessary.

Response from the authors: Thank you for your comment. We deleted “in order”.

Comment: Lines 80-81, “reduce the population's health indicators” is not clear.

Response from the authors: Thank you for your comment. We rewrote the sentence to make it clearer.

Comment: Line 117, remove the first “the.”

Response from the authors: Thank you for your comment. We removed the first “the”.

Comment: Line 137, suggest capitalizing “Indigenous.”

Response from the authors: Thank you for your suggestion. We capitalize “Indigenous”.

Comment: Line 169, “adherence to family health,” change the “to” to “of”

Response from the authors: Thank you for your comment. We made that change.

Comment: Line 210: “other NASF professionals nonphysicians,” this is not clear. Also need to spell out acronym.

Response from the authors: Thank you for your comment. We rewrote the sentence to make it clearer.

Comment: Review manuscript for consistency in capitalization of “Regions” when referring to a specific numbered region. They should all be capitalized.

Response from the authors: Thank you for your comment. We made all the capitalized to refer to specific “Region”.

---

## [Editor Report · Decision Letter 2]

21 Oct 2020

Performance of family health teams for tackling chronic diseases in a state of the Amazon

PONE-D-19-35881R2

Dear Dr. Gomes Alves,

We’re pleased to inform you that your manuscript has been judged scientifically suitable for publication and will be formally accepted for publication once it meets all outstanding technical requirements.

Kind regards,

Nelly Oelke

Academic Editor

PLOS ONE

Additional Editor Comments (optional): Thank you for making the final revisions to your manuscript. 
---

## [Editor Report · Acceptance letter]

27 Oct 2020

PONE-D-19-35881R2 

Performance of family health teams for tackling chronic diseases in a state of the Amazon. 

Dear Dr. Alves:

I'm pleased to inform you that your manuscript has been deemed suitable for publication in PLOS ONE. Congratulations! Your manuscript is now with our production department. 

Kind regards, 

on behalf of

Dr. Nelly Oelke 

Academic Editor

PLOS ONE